# Bioinspired mechanical mineralization of organogels

Jorge Ayarza[1,6], Jun Wang[1,6], Hojin Kim [1,2], Pin-Ruei Huang[1], Britteny Cassaidy[1], Gangbin Yan [1], Chong Liu [1], Heinrich M. Jaeger[2,3], Stuart J. Rowan [1,4,5] & Aaron P. Esser-Kahn [1] ✉

Mineralization is a long-lasting method commonly used by biological materials to selectively strengthen in response to site specific mechanical stress. Achieving a similar form of toughening in synthetic polymer composites remains challenging. In previous work, we developed methods to promote chemical reactions via the piezoelectrochemical effect with mechanical responses of inorganic, ZnO nanoparticles. Herein, we report a distinct example of a mechanically-mediated reaction in which the spherical ZnO nanoparticles react themselves leading to the formation of microrods composed of a Zn/S mineral inside an organogel. The microrods can be used to selectively create mineral deposits within the material resulting in the strengthening of the overall resulting composite.

Mineralization is a strategy commonly used in biology to provide mechanical resistance and structural support to tissue. Moreover, the morphology of the mineral growth is tuned to form hierarchical structures with diverse levels of complexity[1-4] Biomaterials such as bone, shells, and exoskeletons display great adaptation to the environment by responding to mechanical stress and generating mineralized structures with optimized mechanical responses[2,5-9]. These biological systems have inspired several examples in the literature of synthetic materials with similar adaptability. For example, the growth of calcium minerals (i.e., hydroxyapatite) in hydrogel matrices has been largely studied for applications in tissue engineering and regenerative medicine[1,10-18]. Moreover, the combination of synthetic methodologies with advanced manufacturing techniques (3D printing) has yielded mineralized hydrogels with complex architectures[19,20]. Only one, excellent example of mechanically mediated mineralization via the piezoelectric effect exists from the Kang group[21]. In that work, the mechanical loading of a piezoelectric polymer scaffold promoted the mineralization of apatite from surrounding media onto its surface. This simple method allowed for the fabrication of structured composites by controlling the distribution of mechanical stress along the scaffold. To our knowledge, very few mineralization methodologies exist in synthetic composite materials and organic solvents. Previous examples include the growth of metal or inorganic nanoparticles in organogels and the crosslinking of polyethyleneimine organogels via fixation of $CO_2$ to form carbamate salts[22-24]. To begin moving the exciting properties of biology from aqueous systems to the manufacturing realms of extrusion and composite fabrication, we sought a method of mineralization in synthetic systems, wherein we could in situ fabricate minerals within a polymer matrix to access composites with tailored mechanical properties.

In previous work, our group and others have explored the use of piezoelectric nanoparticles to conduct mechanically-promoted polymerization reactions by harnessing the piezoelectrochemical effect[25-29]. In recent work, we observed that ZnO nanoparticles promoted the thiol-ene and thiol-disulfide reactions, under the effect of controlled mechanical vibrations or ultrasound[30,31]. In exploring this reactivity further, we discovered that ZnO nanoparticles reacted with 2-mercapto-5-methyl-1,3,4-thiadiazole (McMT) under mechanical stimulation in an organic solution to

[1]Pritzker School of Molecular Engineering, University of Chicago, 5640 South Ellis Avenue, Chicago, IL 60637, USA. [2]James Franck Institute, University of Chicago, 929 East 57th Street, Chicago, IL 60637, USA. [3]Department of Physics, University of Chicago, 5720 South Ellis Avenue, Chicago, IL 60637, USA. [4]Department of Chemistry, University of Chicago, 5735 South Ellis Avenue, Chicago, IL 60637, USA. [5]Chemical and Engineering Sciences Division, Argonne National Laboratory, 9700 Cass Avenue, Lemont, IL 60439, USA. [6]These authors contributed equally: Jorge Ayarza, Jun Wang. ✉e-mail: aesserkahn@uchicago.edu

form rod-shaped crystalline microparticles (microrods)[32–34]. After the characterization of the mineralization process, we envisioned that such a reaction could be used for mineralizing polymer organogels, thereby strengthening them.

In this paper, we present the inorganic mineralization of polymer composites using mechanically induced formation of crystalline microrods from the partial consumption of ZnO nanoparticles via reaction with an energetically loaded thiol−McMT. We study the formation of the microrods, characterize their chemical composition and potential routes for formation, and assess their rheological properties in suspension. This mineralization process can be used to alter the mechanical properties of piezo-responsive composite materials thus creating materials that spontaneously induce mineralization to achieve mechanically mediated hardening. Unlike the methods previously described in the literature, our strategy allows for the bulk transformation of spherical, piezoelectric nanoparticles into rod-shaped microparticles within the polymer matrix, where the mineral growth is directed by the concentration of mechanical stress, albeit at the cost of precise control over the mineral deposition. Additionally, it can be compatible with various polymer substrates and solvents as long as the polymerization chemistry is orthogonal to the synthesis of the microrods.

## Results and discussion

### Synthesis of the microrods and chemical characterization

Previously, our group demonstrated that materials containing ZnO nanoparticles ($\Phi$ = 18 nm) can generate reactive thiols via mechanical vibration and piezoelectric reduction[30,31]. We proposed that the thiol, which was pre-associated with the ZnO, was activated upon mechanical agitation to generate a thiyl radical. In exploring this reaction further, we employed a redox-sensitive thiol, 2-mercapto-5-methyl-1,3,4-thiadiazole (McMT). The reaction yielded the spontaneous formation of rod-shaped microparticles (microrods) which transformed the reaction mixture into a slurry after sonication (40 kHz) for 6 h (Fig. 1a).

In initial experiments, a solution of McMT in dimethylformamide (DMF, 1.25 M) was prepared and ZnO nanoparticles (5 wt%) were added. The nanoparticles were homogenously dispersed, and the mixture was sonicated in an ultrasound bath (40 kHz) for 6 h. To characterize the product, it was diluted with methanol, and the precipitate was recovered by centrifugation (3000 x $g$). SEM imaging of the precipitate revealed the presence of crystalline microrods with an average length of 4.3 (±1.2) µm and an average width of 0.4 (±0.1) µm (Fig. 1b). By visual inspection, the microrods displayed insolubility in water and most organic solvents, except for polar, aprotic solvents

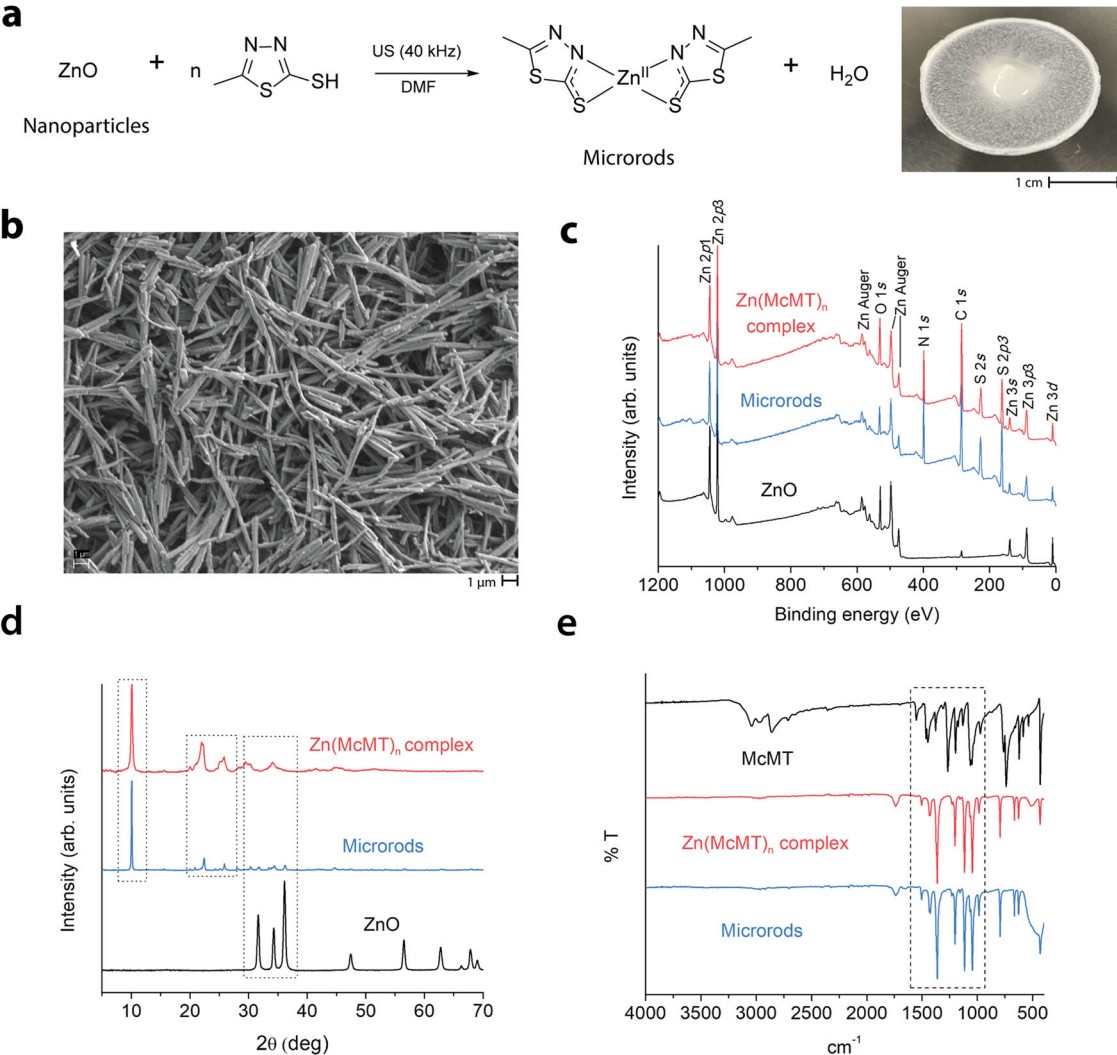

**Fig. 1 | Synthesis and chemical characterization of the microrods. a** Plausible reaction scheme showing the formation of the coordination compound Zn(McMT)$_n$ and picture of white slurry consisting of a microrod suspension in DMF; **b** SEM image of the microrods (scale bar 1 µm); **c** XPS and **d** XRD spectra of ZnO nanoparticles, microrods, and Zn(McMT)$_n$ complex; and **e** FTIR spectra of McMT, Zn(McMT)$_n$ complex, and microrods.

such as dimethylformamide and dimethyl sulfoxide under heat (70 °C)[32]. Of note, the microrods easily dissolved in a solution of the reducing agent tris(2-carboxyethyl)phosphine hydrochloride (TCEP) in DMF. Previous studies in the literature have shown that TCEP can dissolve metal-organic complexes through ligand substitution to form a soluble chelate[35]. Moreover, McMT does not form rods upon redox reactions on its own. Based on these observations and on literature precedent for reactions between McMT and Zn, we conjectured that the thiol reacted with the ZnO nanoparticles to form a coordination compound (Fig. 1a)[30,31,33,36–38]. We hypothesize that the vibration energy may promote the release of the coordinated $Zn^{2+}$ cations from the nanoparticles, as well as facilitate the growth of the crystals.

To determine the chemical composition of the microrods, we used X-ray photoelectron spectroscopy (XPS), scanning transmission electron microscopy with energy dispersive X-ray spectroscopy (STEM-EDS), combustion elemental analysis, powder X-ray diffraction (pXRD), and infrared spectroscopy (FTIR). Previous literature has shown that the coordination compound of McMT with $Zn^{2+}$ ions yields an insoluble crystalline material with a suggested composition of $ZnL_2$, although no crystal structure was reported[33]. We synthesized the $Zn(McMT)_n$ complex by adapting a method from the literature[33]— mixing soluble zinc nitrate, $Zn(NO_3)_2 \cdot 6H_2O$, with McMT in a basic aqueous solution (Supplementary Information Section 1). The resulting insoluble, white precipitate was compared with the microrods obtained from the sonication experiment. The XPS spectrum of the microrods (Fig. 1c) showed the characteristic peaks of Zn 2$p$, O 1$s$, N 1$s$, C 1$s$, and S 2$s$, and matched the spectrum of the $Zn(McMT)_n$ complex.

Of note, the O 1$s$ peak shifted from 530 eV in ZnO to 532 eV in the microrods (Supplementary Fig. 1). This result suggests that the microrods no longer contain crystalline ZnO, but have small amounts of water or hydroxide[39]. The STEM-EDS images showed an even distribution of Zn, N, and S in both samples (Supplementary Figs. 2 and 3). The abundance of oxygen was considerably less when compared to the other elements. The elemental analysis showed a similar abundance of C, H, N, and S in both samples, with a suggested composition of Zn C6 H8 N4 S4 (Supplementary Figs. 4 and 5), supporting the proposed composition from the literature of $Zn(McMT)_2$. The XRD spectrum of the microrods (Fig. 1d) also matched the spectrum of the $Zn(McMT)_n$ complex, although it also contained phases of the ZnO in a very small proportion. These peaks can be attributed to unseparated ZnO particles trapped within the microrods from the original reaction. Finally, the FTIR spectrum (Fig. 1e) and modulated differential scanning calorimetry (DSC) curve of the microrods (Supplementary Figs. 6–8) were also in accordance with the previous results. Collectively, these results indicate that the microrods are composed of a coordination compound between $Zn^{2+}$ and McMT. Unfortunately, due to issues with low solubility, it was challenging to obtain a crystal of sufficient size to conduct single crystal XRD and thus confirm the exact composition of the complex and that the ratio is most likely 1:2 of Zn:McMT. While the mechanism of formation is not definitive, we propose the following based on our previous experiments. First, the thiol forms a transient bond with the surface of the ZnO nanoparticles. We have confirmed this interaction in our previous work, and it has been observed by others using adsorption experiments[30,31,40]. After this bond occurs,

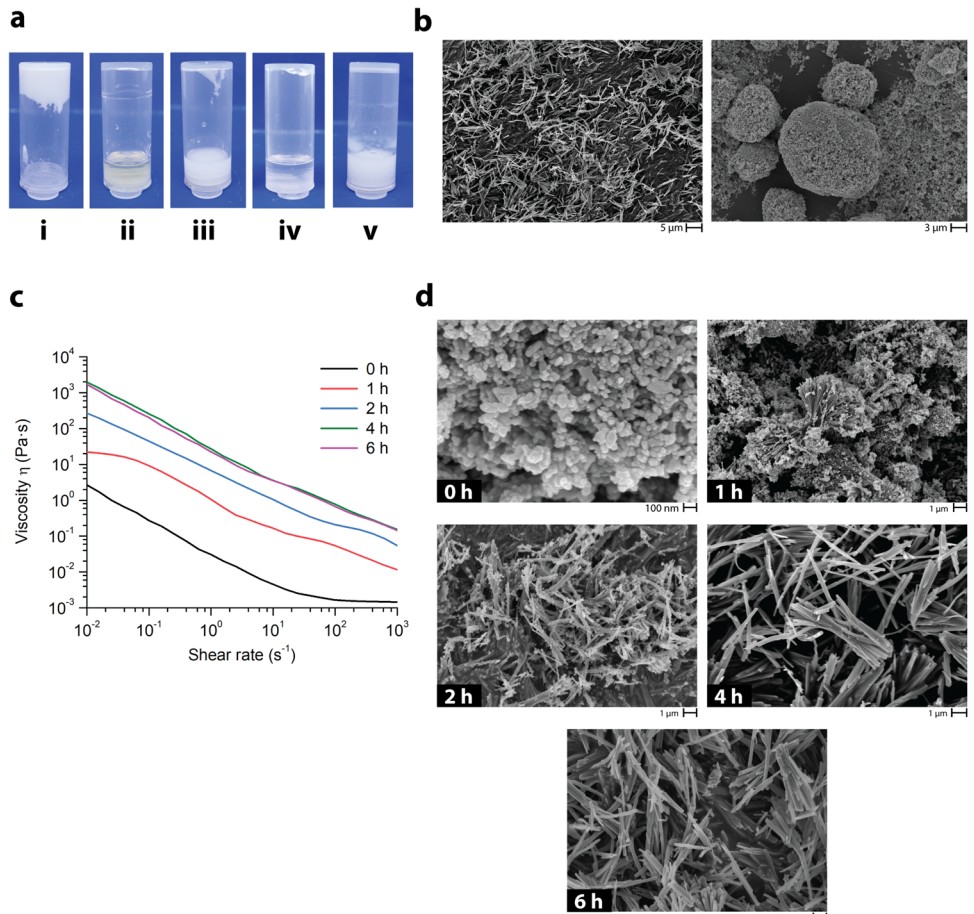

**Fig. 2 | Study of the growth of the microrods under sonication (US 40 kHz). a** Photos of control reactions: (i) all components, (ii) no ZnO, (iii) no US (vortex 400 rpm), (iv) w/ ZnBr2, and (v) w/ McMT disulfide dimer. **b** SEM images comparing the products of the reaction with (*left*) and without (*right*) ultrasound. **c** Shear viscosity measurements of the formation of the microrods at different reaction timepoints. **d** SEM images of the products isolated at different reaction timepoints.

under the effect of vibration, the McMT, due either to its electronics or the presence of the adjacent N, reacts and sequesters a $Zn^{2+}$ ions to form the complex, which then crystallizes in situ initiating the formation of the crystalline microrods.

## Study of the formation and growth of the microrods

To investigate the formation of the microrods in more detail, we conducted several control experiments (Fig. 2a) that explored the effect of the reagents and reaction conditions. In the absence of ZnO nanoparticles, no microrods were formed. Replacing the ultrasound with mechanical stirring using a magnetic stir bar (400 rpm) resulted in the formation of microrods with similar dimensions but at a significantly lower yield, 40 % (Supplementary Fig. 9). Simple vortex agitation (400 rpm) of the reaction mixture resulted in large particle aggregates with uneven shapes (Fig. 2b). When the ZnO nanoparticles were replaced with a soluble, pH-neutral zinc salt ($ZnBr_2$), no microrods were formed. When the McMT was replaced with its disulfide dimer, thus removing the reactive thiol, again no microrods were formed (Fig. 2a). Together these results are consistent with the vibration-promoted growth of the microrods from the ZnO nanoparticles and the presence of a reactive thiol.

Next, we sought to determine the optimal reaction conditions for the growth of the microrods by varying the stoichiometric ratio between McMT and ZnO nanoparticles. Based on the previous results, we hypothesized a stoichiometry of 1:2 of Zn:McMT was needed to perpetuate rod growth. In each experiment, the rheological properties of the slurry formed under sonication were investigated before the resulting product was washed, dried, weighed, and imaged via SEM. In the first set of experiments, the concentration of McMT was held constant (0.50 mmol in 400 μL of DMF) and the amount of ZnO was varied from 10, 20, and 40 mg, which corresponded to a $Zn^{2+}$ molar equivalent of 0.13, 0.25, and 0.50 mmol, respectively (Supplementary Table 1, Supplementary Fig. 10). In another set of experiments, the amount of ZnO was held constant (20 mg, 0.25 mmol [Zn]) and the concentration of McMT was varied from 0.25, 0.50, and 1.00 mmol in 400 μL of DMF (Supplementary Table 2 and Supplementary Fig. 14). In both sets of experiments, the highest yield of microrods was approximately 80 %. No significant increase in yield was observed when there was a stoichiometric excess of either reactant. Additionally, the highest viscosity of the resulting slurry was obtained when the molar ratio of ZnO:McMT was 1:2. SEM images confirmed that better microrod morphology was obtained under such conditions (Supplementary Figs. 11–13, 15–17). As such, these experiments provided improved conditions for microrod formation.

To gain a better understanding of the growth of the microrods, we measured the shear viscosity (Pa·s) of the slurry at different timepoints. As shown in the graph in Fig. 2c, the viscosity of the slurry varied from ca. 1 Pa·s at 0 h to 1000's Pa·s at 6 h, reaching a maximum between 4 and 6 h of sonication time. The product at each timepoint was isolated and characterized via SEM (Fig. 2d) and particle size analysis (Supplementary Table 3). SEM showed that at 0 h the solid consisted of the spherical ZnO nanoparticles. At the 1 and 2 h timepoints, the images showed a mixture of round- and rod-shaped particles (Supplementary Figs. 18 and 19). Based on the proposed mechanism, we hypothesize that the ZnO nanoparticles act as nucleation sites for the growth of the microrods, which would also explain the presence of ZnO phases in the XRD spectrum of the microrods. Finally, at the 4 and 6 h timepoints, microrod structures of comparable size were predominantly observed.

## In situ growth of the microrods in polymer composites

To show the potential use of the mechanically-promoted mineralization method to reinforce polymeric materials, we grew the microrods within (a) a polymer solution in an organic solvent and (b) an organogel, testing the appropriate mechanical properties in each

case. In the first experiment, a viscoelastic polyurethane polymer solution was used as a matrix owing to its traditional application for the production of plastics and adhesives. Briefly, poly(propylene glycol) tolylene 2,4-diisocyanate terminated ($M_n$ 2300 Da), tetraethylene glycol, McMT, ZnO nanoparticles, dibutyltin dilaurate, and DMF were mixed together. The molar ratio of OH:NCO was set to 1:2, so that a polymer mixture would form. Then, the mixture was sonicated (40 kHz) for 6 h. Additionally, control reactions with either no McMT or no ZnO nanoparticles were run simultaneously. Light microscopy and SEM images confirmed the formation of the microrods within the polymer solution (Supplementary Figs. 20 and 21). Compared to the microrods grown in plain solution, those grown within the polymer solution were 74 % longer and 50 % thicker, with an average length = 7.5 (±3.2) μm and width = 0.6 (±0.2) μm (Supplementary Fig. 22). As shown in Fig. 3, the formation of microrods increased the low-shear viscosity of the polymer solution ($\eta$ = 100 Pa·s) and resulted in shear-thinning behavior. By contrast, the control

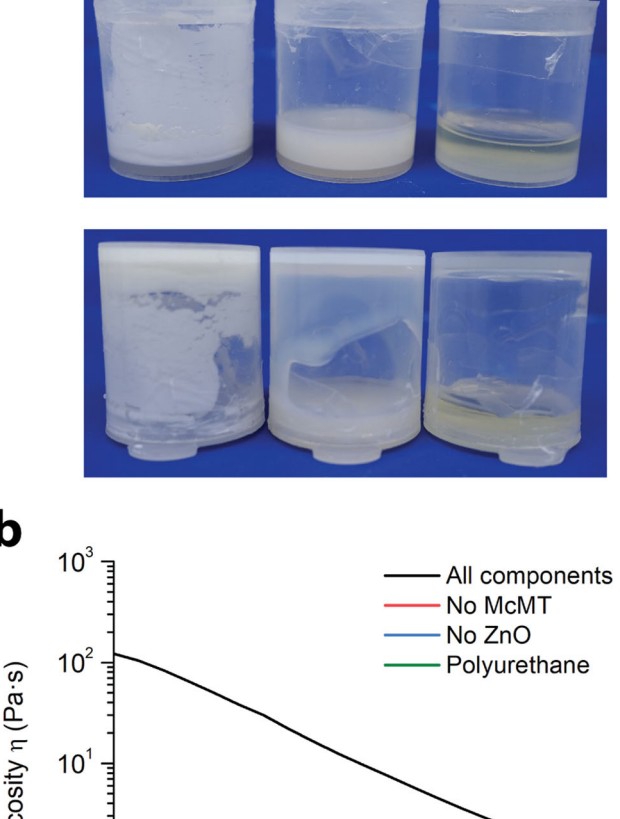

**Fig. 3 | Modifying the rheology of a polymer solution by growing microrods in situ. a** Photos of representative samples (*left to right*): All components, No McMT control, and No ZnO control. After the formation of the microrods, the viscosity of the solution changes, thus preventing flow when the vial is turned upside down. This phenomenon does not occur in the control samples. **b** Shear viscosity measurements of the samples. The shear viscosity measurement of the polyurethane alone is included for reference. All reactions and rheological measurements were done in triplicate (Supplementary Figs. 23 and 24).

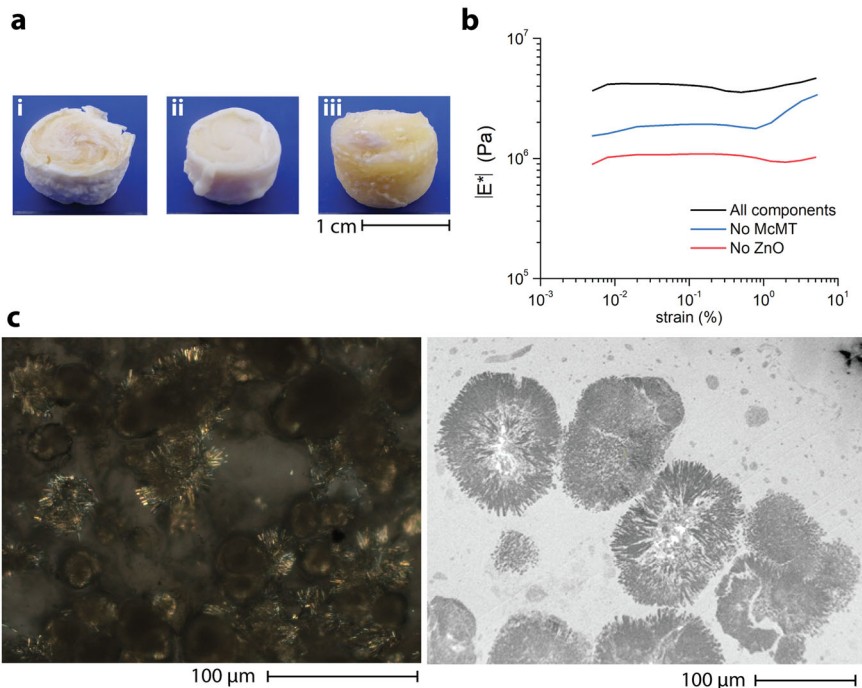

**Fig. 4 | In situ growth of the microrods within a crosslinked polymer organogel.** **a** Photos of the resulting polymer composites after sonication and drying: (i) All components, (ii) No McMT control, and (iii) No ZnO control. **b** Representative DMA oscillatory amplitude measurements for the polymer composite samples. **c** Light microscopy with cross-polarizer (*left*) and TEM (*right*) images of the *All components* sample showing the presence of the crystalline microrods in bundles.

samples exhibited approximately Newtonian behavior with a viscosity ($\eta$) of 3 Pa·s.

In biological systems, mineralization on a scaffold can lead to the formation of ultra-tough materials with unique properties. Thus, we explored the use of a crosslinked polymer organogel as the matrix for the growth of the microrods. To prevent cross-reactivity between the polymer matrix and McMT, we relied on a strain-promoted azide-alkyne cycloaddition crosslinking reaction (Supplementary Fig. 25). To generate the matrix, a linear polyurethane with azide side-groups (azido-PU, $M_n$ = 21.6 kDa, $Đ$ = 1.5) was synthesized using conventional base-catalyzed polymerization between diisocyanates and dialcohols. Then, in a 5 mL plastic vial, a solution of azido-PU and McMT in DMF was prepared. Subsequently, ZnO nanoparticles were added and homogenously dispersed by sonicating the mixture for 30 s followed by vortex stirring for 2 min. The mixture was cooled down to approximately 4 °C for 20 min. Separately, a crosslinker solution of bis(dibenzocyclooctyne-amine) terminated pentaethylene glycol (DBCO-PEG$_5$-DBCO) 0.35 M in DMF was prepared. The crosslinker solution was rapidly added to the polymer mixture and stirred vigorously in a vortex for 2 min. The organogel formed within minutes. It was then sonicated in an ultrasound bath (40 kHz) for 6 h. The gels were taken out from the vials and dried under vacuum at 60 °C overnight. For comparison purposes, control samples without either McMT or ZnO nanoparticles were also prepared (Fig. 4a).

The composites were tested with dynamic mechanical analysis (DMA) using amplitude and frequency sweeps under compression mode. The measurements were done in triplicate for each control,

with a new sample in each measurement (Table 1 and Supplementary Fig. 26). As shown in Fig. 4b, the composite with both McMT and ZnO displayed a considerably higher complex modulus ($|E^*|$) in the linear region than the control samples without either McMT or ZnO. The frequency sweep measurements confirmed that all samples behaved as crosslinked polymers. Additionally, as shown in Table 1, the composite containing all the components had a complex modulus of nearly twice that of the composite without McMT, and more than three times the complex modulus of the composite without ZnO. We attributed the increase in modulus to the growth of the microrods within the matrix of the composite. Moreover, we tested how different loadings of the starting materials, and hence the microrods, affected the modulus of the resulting composite (Supplementary Figs. 27 and 28). We loaded the azido-PU gel with different amounts of McMT and ZnO (2X, 1X, and 0.5X). For reference, 1X corresponds to the conditions reported originally. Our results showed an approximately fourfold increase in the elastic modulus of the composite when we doubled the amount of ZnO and McMT. However, when we reduced those amounts tso half, the resulting modulus was comparable to the original conditions. To confirm the presence of the microrods, a piece of the composite was imaged using light microscopy (Supplementary Fig. 29). To obtain a better image, the piece was swollen with a drop of tetrahydrofuran (THF) and the cross-polarizer filter confirmed the presence of bundles of crystalline microrods embedded in the polymer matrix (Fig. 4c). TEM images of cross-sections of the composites confirmed what was observed under light microscopy (Fig. 4c and Supplementary Fig. 30).

In conclusion, we demonstrate a method to synthesize crystalline microrods from the reaction of ZnO nanoparticles and McMT using ultrasound agitation. Chemical characterization of the microrods strongly supported that the composition was a Zn(McMT)$_n$ complex in a crystalline form. We studied the formation of the microrods using rheology and SEM imaging. The results suggested that the ZnO nanoparticles served as both the source of Zn$^{2+}$ ions and acted as nucleation sites for the growth of the crystals. Moreover, we showed two examples of potential applications of the microrods by growing

**Table 1 | Statistical analysis of the complex modulus DMA measurements for the polymer composites**

| Sample | $|E^*|$ (MPa) | s (MPa) |
|---|---|---|
| All components | 3.47 | 0.37 |
| No McMT | 1.65 | 0.35 |
| No ZnO | 1.01 | 0.04 |

them within a polymer solution and an organogel, thus emulating the process of biomineralization in a synthetic organic environment. Our future work will focus on more detailed studies of the growth of these microrods and controlling their morphology under different conditions such as continuous shear stress or using our electrodynamic shaker setup. To improve the mechanical properties of the composite, it may also be possible to improve the bonding between the polymer matrix and the microrods.

## Methods

### Representative synthesis of the microrods
In a cylindrical polypropylene vial, McMT (66 mg, 0.50 mmol) was dissolved in DMF (400 µL). ZnO nanoparticles (20 mg, 0.25 mmol) were added to the solution and homogenously dispersed via sonication for 20 s. The mixture was sonicated (40 kHz) in the dark for 4 h. The product was diluted with methanol (5 mL) and separated by centrifugation (3000 x $g$ for 10 min). The precipitate was washed twice with methanol (10 mL) and again separated by centrifugation. The product was dried at 50 °C under vacuum overnight. Yield: 71 mg, 86 %.

### Synthesis of Zn(McMT)$_n$ complex
A solution of Zn(NO$_3$)$_2$·6H$_2$O (149 mg, 0.50 mmol) in water (1 mL) was prepared. In a separate vial, McMT (132 mg, 1.00 mmol) was dissolved in a NaOH$_{(aq)}$ 1 M solution (1 mL). The zinc nitrate solution was added dropwise to the McMT solution under vigorous stirring. A white precipitate formed, and the mixture was stirred for 30 min. The solid was recovered by filtration and washed thoroughly with water. The product was dried at 50 °C under vacuum overnight. Yield: 128 mg, 78 %.

### Synthesis of the microrods within polymer organic solution
In a cylindrical polypropylene vial, poly(propylene glycol) tolylene 2,4-diisocyanate terminated (PPG diisocyanate, M$_n$ 2300 Da, 3.60 w % NCO, 0.50 g, 0.43 mmol NCO) was weighed. PPG diisocyanate was dissolved in DMF (1 mL) and 21 mg of tetraethylene glycol (0.11 mmol glycerol, 0.22 mmol OH) were added. Dibutyltin dilaurate (3 µL) was added to the mixture and the solution was vortexed for 30 s. Then, McMT (132 mg, 1.00 mmol) was added, and the mixture was vortexed until dissolution. Finally, ZnO nanoparticles (40 mg, 0.50 mmol) were added and dispersed under sonication. The mixture was sonicated in an ultrasound bath (40 kHz) for 6 h.

### Synthesis of polymer organogel with embedded microrods
In a cylindrical polypropylene vial, azido-PU (150 mg) was dissolved in DMF (400 µL) overnight. Separately, in a vial, DBCO-PEG$_5$-DBCO (15 mg) was dissolved in DMF (50 µL). McMT (66 mg) was added to the azido-PU solution and dissolved by vigorous stirring. ZnO nanoparticles (20 mg) were added and homogenously dispersed by vigorous stirring and sonication for 20 s. The mixture was cooled down in a fridge (4 °C) for 30 min. The crosslinker solution was quickly added to the azido-PU mixture and rapidly vortexed. The organogel forms within 1 min. It was then sonicated in an ultrasound bath (40 kHz) for 6 h. The organogel was then carefully removed from the vial and dried in a vacuum oven (50 °C) overnight to remove the DMF.

## Data availability
The synthesis, characterization, and experimental data for this paper are all provided in the Supplementary Information. All other data is available from the corresponding author upon request.

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

## Acknowledgements

This work is supported by the US Air Force Office of Scientific Research (AFOSR) through Grant COE 5-29168, the National Science Foundation (NSF) through Grant CHE-1710116, the US Army Research Office (ARO) through Grant W911NF-17-1-0598 (71524-CH), and the University of Chicago Materials Research Science and Engineering Center (MRSEC), which is funded by the NSF under award number DMR-2011854. Parts of this work were carried out at the Soft Matter Characterization Facility (SMCF), Department of Chemistry, and Materials Research Science and Engineering Center (MRSEC) facilities of the University of Chicago. We thank Dr. Alexander Filatov, manager of the University of Chicago X-ray research facilities for providing technical support in XPS and XRD measurements. We thank the University of Chicago Advanced Electron Microscopy Core Facility (RRID:SCR_019198) for providing TEM images of selected samples. We thank the University of Illinois at Chicago for providing access to their Electron Microscopy Core facilities. We thank Dr. Philip J. Griffin, director of the SMCF, for helpful discussions, and Dr. Adam Weiss for help with NMR spectroscopy.

## Author contributions

J.A. and J.W. contributed equally to this work. J.A. and A.P.E.-K. directed the project and wrote the manuscript. J.A. and J.W. performed the main experiments and analyzed the data. H.K. contributed to light microscopy imaging of polymer composite samples. P.-R.H. contributed to performing select experiments. B.C. contributed with SEM imaging of the microrods. G.Y. contributed to STEM-EDS imaging of the microrods. S.J.R., C.L., and H.M.J. contributed by providing critical experimental suggestions and reviewing the manuscript. All authors discussed the results and commented on the manuscript.

## Competing interests

The authors declare no competing interests.
