## [Peer Review File · Nature Communications]

Reviewers' Comments:

Reviewer #1:

Remarks to the Author:

This manuscript described the synthesis of microrods, $(\text{Zn}(\text{McMT})_n)$, using ZnO nanoparticles and McMT as precursors in DMF solution. The authors further explored the in situ formation of such rods within polymer networks/cross-linked polymers. The authors highlighted a lot about biomineralization. However, I really don't think this study involves too much biomineralization because biomineralization emphasizes the role of organic matrix/additives on the formation of mineral in terms of size, morphology and structure, etc. In this manuscript, it seems to me microrods form with or without polymers. In other words, the microrods just grow within the polymer networks. It has not shown that there is any intimate interaction/interplay between the microrods and the polymers. How polymers affect the formation of the microrods is unclear. In addition, it's not something new/outstanding that introducing rods-like additives into the polymers increases their shear viscosity/modulus. Therefore, I don't think the scientific importance and novelty of this manuscript warrant its publication in Nat. Commun.

My additional comments are listed below:

- 1, The TOC is confusing; It's difficult to get the point;
- 2, The XPS spectra of ZnO nanoparticles and $\text{Zn}(\text{McMT})_n$ complex should be provided in Figure 1c. Binding energy might shift for ZnO nanoparticles and microrods;
- 3, How microrods loading affects the modulus of the resulting composites? TOC mentioned but it cannot be found in the main text;
- 4, Scale bar should be provided in Figures 1a and 4a.
- 5, High resolution TEM study is recommended and SAED pattern might provide useful information on the structure of the obtained microrods; in addition, how about using high resolution mass spectroscopy and elemental analysis to determine the composition of the $\text{Zn}(\text{McMT})_n$ microrods;
- 6, Is there any difference in terms of morphology and size between the microrods obtained in the presence and absence of the polymers?
- 7, "Fig. 5c" should be "Fig. 4c" in page 10;
- 8, Shear viscosity of polymer alone should be added for comparison purpose.

Reviewer #2:

Remarks to the Author:

The authors present the ultrasound induced mineralization of a synthetic organogel. The work is noteworthy, as mechanically stimulated mineralization is critical to the development of important biomaterials, but it is barely known in synthetic material systems. The key advance in my opinion is the mechanical stimulus, which distinguishes the work from other synthetic mineralization strategies.

One example of related work is ref. 21, cited by the authors. That work also showed piezo-mediated mineralization, but on the surface of the piezoelectric substrate rather than via dissolution of the piezoelectric material. There are advantages and disadvantages to both approaches, largely centered around the question of what you want your bulk mechanical material "host" to be and how much variability you would like (advantages of the present system) vs. using structured materials to direct the growth rather than relying on mechanical stress concentration alone (potential disadvantages of the present system). A stronger presentation/demonstration of these and other differences relative to ref. 21 would strengthen the novelty of this paper.

The actual chemical reaction involved seems relatively well described, although it would be important to comment on (and hopefully provide experimental evidence for) the fate of the oxygen from ZnO. Also, the authors report the best yields at 2 eq. McMT per Zn atom, but this only makes sense if all of the Zn is converted from particles to mineral rods. Is this the case? And why would excess McMT reduce the efficiency of conversion (the authors state 1:2 is better than 1:4).

My biggest question is about the nature of the mechanical stimulation. Could agitation, rather than mechanical stimulation, be the main factor? It is not clear how mixing compares in the stirring vs. sonication cases. What of vortex mixing vs. stirring? The authors state that there is a load-dependent response, but I do not know how to quantify the load in sonication vs. stirring. Is "load" really the right term? The TOC graphic seems a bit misleading, since I do not see static loading in the paper.

If all of the ZnO converts, or close to it, is it likely that a piezo response is involved? At some point, do the particles cease to be sufficiently piezo-active for that to propagate the mechanism?

With the above points addressed, I believe the work is likely to merit publication in NCOMMS.

REVIEWER #1:

GENERAL COMMENT. “This manuscript described the synthesis of microrods, (Zn(McMT)n), using ZnO nanoparticles and McMT as precursors in DMF solution. The authors further explored the in-situ formation of such rods within polymer networks/cross-linked polymers. The authors highlighted a lot about biomineralization. However, I really don’t think this study involves too much biomineralization because biomineralization emphasizes the role of organic matrix/additives on the formation of minerals in terms of size, morphology and structure, etc. In this manuscript, it seems to me microrods form with or without polymers. In other words, the microrods just grow within the polymer networks. It has not shown that there is any intimate interaction/interplay between the microrods and the polymers. How polymers affect the formation of the microrods is unclear. In addition, it’s not something new/outstanding that introducing rods-like additives into the polymers increases their shear viscosity/modulus. Therefore, I don’t think the scientific importance and novelty of this manuscript warrant its publication in Nat. Commun.”

Author reply: We would like to thank this reviewer for their comments. We regret that our current presentation of this work was not sufficiently clear. We believe our mistake was in not clarifying that we were trying to create a bio-inspired process analogous to biomineralization. Not biomineralization itself. We agree that introducing rod-like additives is not novel. We want to emphasize that the novelty is in the mechanical activation of the rod growth from an inorganic and organic substrate. This allows the formation of these rods within the material only upon mechanical activation of the material.

Our intent was to highlight the phenomenon of biomineralization as an example of autonomous material remodeling in biological systems such as bone, shells, and exoskeletons. Our group has taken inspiration from such systems to develop methods for conferring remodeling properties to synthetic composites with the intention of developing advanced materials with better durability and adaptability. We recognize that perhaps the introduction of the paper is an over-extension of this idea and may confuse the reader because the new chemistry described herein is not of comparable complexity and tunability as that of biomineralization in cellular systems. To address this issue, we have revised the introduction and discussion of the relevant results to make a clear distinction between our work and the concept of biomineralization.

The reviewer points out that the microrod formation can occur in the absence of the polymer matrix, that there is no evidence of direct interaction between the microrods and the polymers, and that the addition of rod-shaped additives into polymers to change their modulus/viscosity is not outstanding or novel. In reference to the points above we would emphasize the following:

- In this work the microrods do not grow out of a solution as in metal-organic synthetic methodology, but rather there is an *in situ* conversion of spherical ZnO nanoparticles to rod-shaped metal-organic microparticles within a polymer matrix.

- Our experiments were intended to show that the polymer matrix alters the size and morphology of the rods. Therefore, the polymer, while not directly interacting, does influence both the size of the rods and subsequently the mechanical properties they enable. There is also literature precedent for mineralization within hydrogels with calcium phosphate mineral that do not directly rely on a direct interaction between the inorganic phase and the polymer strands. Finally, we are eager to examine a polymer system with direct interaction with the microrods but expect this to be part of future publications.
- Careful and extensive studies (not shown in the manuscript) were conducted to identify the polymer systems used in the paper that are chemically compatible with the microrod formation. We also selected only chemistries that are compatible with commercial processes e.g., polyurethanes. Hence, we envision that the microrods could be incorporated on existing, commercial composites.

In summary, we believe that the work presented in this paper describes a novel methodology for mineralization of synthetic polymer composites in organic substrates. We hope this explanation provides further clarity upon the novelty of this process. While there are several methods of mineralization or rod formation that can occur within a material there are very few that are triggered by mechanical energy, and which subsequently act in a self-reinforcing manner.

COMMENT #1. "The TOC is confusing; It's difficult to get the point."

Author reply: We apologize for the confusion around the TOC. Upon further consideration and discussion, we agree and thank the reviewer for this comment. We have updated the TOC in an effort to provide a clearer picture of the main advancement described in this paper.

COMMENT #2. "The XPS spectra of ZnO nanoparticles and Zn(McMT)_n complex should be provided in Figure 1c. Binding energy might shift for ZnO nanoparticles and microrods."

Author reply: We thank the reviewer for the suggestion. We conducted the XPS measurements for ZnO, microrods and Zn(McMT)_n complex (Fig. S1), and updated Figure 1c. The XPS spectra of the microrods and Zn(McMT)_n complex matched. The Zn 2p_{3/2} peak with binding energy 1022 eV confirms the presence of Zn²⁺ in both ZnO and microrods. The O 1s peak in ZnO appears at 530 eV, whereas the O 1s peak in the microrods appears at 532 eV. This shift in binding energy suggests that the O atoms present in the microrods might not be part of the ZnO crystalline structure but rather from other ligands such as water or hydroxide. However, these results are not definite and need to be interpreted with the other data. It does match all the previous values observed for these species.

Reference: *J. Phys. Chem. C* **2020**, 124, 7777–7789.

COMMENT #3. “How microrods loading affects the modulus of the resulting composites? TOC mentioned but it cannot be found in the main text.”

Author reply: We thank the reviewer for the suggestion. To clarify the point, we conducted additional experiments where we loaded the azido-PU gel with different amounts of McMT and ZnO, namely 2X, 1X and 0.5X (Figs. S27,28). For reference, 1X corresponds to the conditions reported originally. Our results showed a considerable increase in the elastic modulus of the composite (approximately 4 times in magnitude) when we doubled the amount of ZnO and McMT. However, when we reduced those amounts to half, the resulting modulus was comparable to the original conditions.

COMMENT #4. “Scale bar should be provided in Figures 1a and 4a.”

Author reply: We thank the reviewer for pointing out the mistake and have included the missing scale bars.

COMMENT #5. “High resolution TEM study is recommended and SAED pattern might provide useful information on the structure of the obtained microrods; in addition, how about using high resolution mass spectroscopy and elemental analysis to determine the composition of the Zn(McMT)_n microrods.”

Author reply: We appreciate the useful suggestions for further chemical characterization of the microrods. We imaged the microrods with STEM and conducted elemental mapping with EDS (Figures S2,3). The images showed a homogenous distribution of Zn, S and N atoms across the microrod structure supporting that they are composed almost entirely of the Zn(McMT)_n material. The abundance of O was considerably less than the other elements supporting that almost all the ZnO has been consumed. After multiple attempts, we were unable to obtain an SAED spectrum of sufficient quality to determine the crystal structure. This may be due to there being more than 1 phase within the rods. To address the question of composition, we conducted elemental analysis of both the microrods and Zn(McMT)_n complex. In both samples, we detected a ratio of Zn, H, N and S atoms that supported the assignment (Figure S4,5) composition of the microrods as Zn(McMT)₂ with the ratios of EA matching to a very high degree.

COMMENT #6. “Is there any difference in terms of morphology and size between the microrods obtained in the presence and absence of the polymers?”

Author reply: We thank the reviewer for the suggestion. To address this point, we took light microscopy and SEM images of the microrods grown within the polyurethane pre-polymer solution. Their morphology was similar to ones grown in solution, however, particle size analysis

showed they were 74% longer and 50% thicker when grown with polymers. We provide these images in the SI (Figs. S20,21) and size distribution graphs (Fig. S22).

COMMENT #7. “Fig. 5c” should be “Fig. 4c” in page 10.

Author reply: We thank the reviewer for pointing out the mistake and we corrected the figure reference in the main text.

COMMENT #8. “Shear viscosity of polymer alone should be added for comparison purpose.”

Author reply: We thank the reviewer for the suggestion. We measured the shear viscosity of the polymer alone and added it to the graph for comparison (Fig. 3b).

REVIEWER #2:

GENERAL COMMENT: “The authors present the ultrasound induced mineralization of a synthetic organogel. The work is noteworthy, as mechanically stimulated mineralization is critical to the development of important biomaterials, but it is barely known in synthetic material systems. The key advance in my opinion is the mechanical stimulus, which distinguishes the work from other synthetic mineralization strategies.”

Author reply: We would like to thank the reviewer for their positive comments about our work and its relevance to the fields of mechanochemistry and materials science. We have assessed the reviewer’s concerns and suggestions below.

COMMENT #1: “One example of related work is ref. 21, cited by the authors. That work also showed piezo-mediated mineralization, but on the surface of the piezoelectric substrate rather than via dissolution of the piezoelectric material. There are advantages and disadvantages to both approaches, largely centered around the question of what you want your bulk mechanical material “host” to be and how much variability you would like (advantages of the present system) vs. using structured materials to direct the growth rather than relying on mechanical stress concentration alone (potential disadvantages of the present system). A stronger presentation/demonstration of these and other differences relative to ref. 21 would strengthen the novelty of this paper.”

Author reply: We appreciate the reviewer’s suggestion about providing a better explanation of the advantages/disadvantages in our system when compared with previous examples in the literature. We have included further comments in the text that we believe highlight the differences per the reviewer’s suggestion.

COMMENT #2: “The actual chemical reaction involved seems relatively well described, although it would be important to comment on (and hopefully provide experimental evidence for) the fate of the oxygen from ZnO. Also, the authors report the best yields at 2 eq. McMT per Zn atom, but this only makes sense if all of the Zn is converted from particles to mineral rods. Is this the case? And why would excess McMT reduce the efficiency of conversion (the authors state 1:2 is better than 1:4).”

Author reply: We thank the reviewer for Further analysis by STEM-EDS and XPS as suggested by Reviewer #1 helped address these questions. The STEM-EDS images showed a considerably lower abundance of O when compared to Zn, N, and S (Figures S2,3), which indicates the consumption of ZnO during the reaction. We theorize the oxygen is converted to other species such as water or hydroxide, as suggested by the shift in the XPS peak for O 1s from 530 eV in ZnO to 532 eV in the microrods. Additionally, some small amount of residual ZnO is detected by

XRD. Unfortunately, since we could not obtain a single crystal XRD spectrum we are unable to confirm the exact final nature of the oxygen species. With regard to the efficiency of the reaction, our results suggest that there is a high conversion (approx. 80 %) of the ZnO to the microrods when the ratio between ZnO:McMT is kept at 1:2. When we tested a 1:2 and 1:4 ratios of ZnO:McMT we obtained nearly identical yields, 65 % and 69 % respectively (Table S2). However, the rheology measurements showed higher viscosity was achieved when the ratio was 1:2 (Figure S14). We theorize that when there is an excess of McMT, some of it might co-crystallize with the microrods thus affecting their growth and the rheological properties of the slurry. We have rephrased the sentences referring to this topic in the main text to clarify the ambiguity.

COMMENT #3: “My biggest question is about the nature of the mechanical stimulation. Could agitation, rather than mechanical stimulation, be the main factor? It is not clear how mixing compares in the stirring vs. sonication cases. What of vortex mixing vs. stirring?”

Author reply: We conducted additional experiments to test the effect of different types of mechanical stimulation (ultrasound, stirring, vortex mixing) on the growth of the microrods (Fig.2 and S9). SEM images of the products showed that only ultrasound and stirring (with a magnetic stir bar) led to the formation of the microrods, whereas vortex mixing led to the formation of clumps. This indicates that the form of mechanical activity directly influences the outcome of the reaction. We plan to investigate this further in future studies and thank the reviewer for the excellent suggestion.

COMMENT #4: “The authors state that there is a load-dependent response, but I do not know how to quantify the load in sonication vs. stirring. Is “load” really the right term?”

Author reply: We appreciate this point and we have corrected the language throughout the manuscript to avoid confusion.

COMMENT #5: “The TOC graphic seems a bit misleading, since I do not see static loading in the paper.”

Author reply: We apologize that the TOC did not clearly convey the purpose and experimental results of the paper, therefore we have updated it.

COMMENT #6: “If all of the ZnO converts, or close to it, is it likely that a piezo response is involved? At some point, do the particles cease to be sufficiently piezo-active for that to

propagate the mechanism?”

Author reply: We thank the reviewer for raising this point. We have not established a direct relationship between the continued growth of the microrods and the piezoelectricity of the remaining ZnO or of the product. We theorize though that piezoelectricity plays a role in the early phase of the process in order to trigger the reaction between the thiol and ZnO, based on our previous studies with thiol reactivity. Our understanding is that the mechanical energy (ultrasound, stirring) contributes to the initial reaction of Zn²⁺ ions from the ZnO nanoparticles. While we cannot confirm that the entire process is driven by piezo-electricity, we have provided sufficient evidence to say that it does not take place unless a piezo-electric event occurs. At this point, it is certainly worth considering this further, but we feel that the use of the piezo term is warranted here.

To test the piezoelectricity of the mixture between the ZnO nanoparticles and growing microrods, we repeated the experiment in Fig. 2c and isolated the insoluble solid product at each timepoint. The product was washed and dried, and then used to prepare a high molecular weight polyethylene glycol composite. This sample was tested for a piezoelectric response using our electrodynamic shaker system coupled with voltage measurement. The results shown below suggest that the microrods themselves are piezoelectric since we observe a considerable increase in the piezoelectric response at later timepoints, in which presumably most of the ZnO has already been consumed. The linearity of the output voltage in response to changes in amplitude further supports the notion that it is a piezoelectric response. Although these results are encouraging, we have not included them in the main text since further experiments are required to explain the observed effect. However, this measurement supports the fact that this process is piezo-electric throughout the reaction.

Procedure for piezoelectric measurements:

For the fabrication of piezo composite, tetra-PEG-NH₂ M_w 20 kDa (50 mg) was dissolved in a DMF/deionized water co-solvent (600 μL volume ratio 1:5) and vortexed well until the solution became clear. The microrods (5 wt %) were then mixed with the above solution via ultrasonication and degassed for 30 min to eliminate the bubbles. The other precursor solution was prepared using tetra-PEG-NHS M_w 20 kDa (50 mg) and the same co-solvent (600 μL) and stored at 4°C for 20 min before mixing to slow the gelation rate. After cooling, the solutions were mixed together and quickly poured into a PTFE mold to cure at room temperature for 30 min. The composite PEG gel (1.50 × 1.25 × 0.20 cm³) was sandwiched between Pt foils serving as electrodes and packed with PET layers for further piezoelectric characterization.

For piezoelectric characterization, the mechanical vibration was generated via an electrodynamic shaker (Modal Shop, model 2025E) with a power amplifier (SmartAmp, model 2100E21–400), while the output voltage was acquired and monitored through a data acquisition module (NI 9234) using SignalExpress 2015 software (National Instruments).

Shaking condition: 200 Hz, Amplitude 0.5 to 2.

Reference: Dong, Yixiao, et al. "Chameleon-inspired strain-accommodating smart skin." *ACS nano* **2019**, 13, 9918-9926.

COMMENT #7: “With the above points addressed, I believe the work is likely to merit publication in NCOMMS.”

Author reply: We appreciate the reviewer’s support for the publication of this manuscript in Nature Communications.

Reviewers' Comments:

Reviewer #1:

Remarks to the Author:

After careful evaluation on the additional experiments and clarification in the revised manuscript, I am happy to recommend this manuscript to be accepted by nat.commun. in its current form.

Reviewer #2:

Remarks to the Author:

The revised manuscript retains the content that made it compelling to me on a first read, and has addressed my concerns. I also appreciated the very constructive suggestions from Reviewer 1, and believe that the paper is further strengthened through the authors' response to those comments.

As a minor point, the authors use the term "novel" on a couple of occasions. This can be a bit of a trigger term, and I do not think that its inclusion is necessary to have the impact of the paper communicated. It is an editorial/author decision, but I would consider removing/replacing the term.

COMMENT # 8: “The revised manuscript retains the content that made it compelling to me on a first read, and has addressed my concerns. I also appreciated the very constructive suggestions from Reviewer 1, and believe that the paper is further strengthened through the authors' response to those comments.

As a minor point, the authors use the term "novel" on a couple of occasions. This can be a bit of a trigger term, and I do not think that its inclusion is necessary to have the impact of the paper communicated. It is an editorial/author decision, but I would consider removing/replacing the term.”

Author reply: We thank the reviewer for their support and suggestions. We have removed the term “novel” from our manuscript in accordance with the Nature Communications guidelines.